# Are Loneliness and Social Isolation Associated with Quality of Life in Older Adults? Insights from Northern and Southern Europe

**DOI:** 10.3390/ijerph17228637

**Published:** 2020-11-20

**Authors:** Giorgi Beridze, Alba Ayala, Oscar Ribeiro, Gloria Fernández-Mayoralas, Carmen Rodríguez-Blázquez, Vicente Rodríguez-Rodríguez, Fermina Rojo-Pérez, Maria João Forjaz, Amaia Calderón-Larrañaga

**Affiliations:** 1Aging Research Center, Department of Neurobiology, Care Sciences and Society, Karolinska Institutet-Stockholm University, 17165 Stockholm, Sweden; amaia.calderon.larranaga@ki.se; 2National School of Public Health, Carlos III Institute of Health, 28029 Madrid, Spain; aayala@est-econ.uc3m.es; 3Health Services Research on Chronic Patients Network (REDISSEC), Carlos III Institute of Health, 28029 Madrid, Spain; jforjaz@isciii.es; 4Center for Health Technology and Services Research (CINTESIS), Department of Education and Psychology, University of Aveiro, 3810-193 Aveiro, Portugal; oribeiro@ua.pt; 5Institute of Economics, Geography and Demography (IEGD), Spanish National Research Council (CSIC), 28037 Madrid, Spain; gloria.fernandezmayoralas@cchs.csic.es (G.F.-M.); vicente.rodriguez@csic.es (V.R.-R.); fermina.rojo@cchs.csic.es (F.R.-P.); 6National Centre of Epidemiology, Carlos III Institute of Health, 28029 Madrid, Spain; crodb@isciii.es; 7The Network Center for Biomedical Research in Neurodegenerative Diseases (CIBERNED), 28031 Madrid, Spain

**Keywords:** aging, loneliness, social isolation, quality of life, SHARE, prospective studies

## Abstract

Purpose: Loneliness and social isolation have detrimental effects on health in old age; however, the prospective associations with quality of life (QoL) remain unclear. Furthermore, despite the existence of a European north-south gradient in the distribution of loneliness and social isolation, little is known whether the associations are context-specific. We investigated the relationships between loneliness, social isolation and QoL of older adults residing in the North (Sweden) and South (Spain) of Europe. Methods: Study sample consisted of 2995 Swedish and 4154 Spanish older adults who participated in waves six and seven of the Study on Health, Aging and Retirement in Europe (SHARE). Loneliness and social isolation were measured at the baseline, and QoL was measured at the baseline and follow-up using CASP-12. Prospective associations were assessed via multivariate linear regression. Results: In Sweden, subjects with higher vs. lower loneliness had 1.01 (95% CI: −1.55, −0.40) units lower QoL, while every standard deviation increase in social isolation was associated with a 0.27 (95% CI: −0.42, −0.09)-unit decrease in QoL. In Spain, every standard deviation increase in social isolation was associated with a 0.66 (95% CI: −1.11, −0.22)-unit decrease in QoL. The association was stronger in subjects aged ≤65 years old and those with no chronic diseases. The association with loneliness was not statistically significant in Spain. Conclusion: Loneliness and social isolation are prospectively associated with decreased QoL among older adults, yet the associations are contextually bound. Future interventions should target both exposures, among others, in order to increase QoL in this group.

## 1. Background

With the significant increase in life expectancy over the last century, more people than ever are expected to reach and live to old age in Europe [1]. The transition into old adulthood is a time of change for many adults. These include retirement, friends and family moving away, loss of societal and gender roles and emotional distress due to death of loved ones, all of which threaten the structural and functional components of their social networks, putting them at risk for loneliness and social isolation.

Loneliness is a negative, subjective experience that is manifested when there is a discrepancy between a person’s actual and desired social needs, while social isolation is the objective state of having few and/or infrequent relationships with others [2]. While sometimes used interchangeably, loneliness and social isolation are not highly correlated with one another [3,4,5], suggesting that older adults can be socially isolated without feeling lonely but also feel lonely despite having a strong and diverse social network.

Loneliness and social isolation have been associated with several negative health outcomes and health-related behaviors, such as mortality, cardiovascular disorders, functional decline and mental disorders [6]. One particular study identified an effect of social isolation on all-cause mortality that rivaled that of traditional risk-factors, such as smoking, alcohol intake, obesity and physical inactivity [7]. However, the association with quality of life (QoL), a person-centered outcome combining elements of an individual’s physical and psychological health, values, beliefs and relationships with each other and the environment, remains unclear [8]. Most empirical evidence comes from non-population-based, cross-sectional studies [9,10,11,12,13]. Moreover, the comparability across studies is hindered by the variability in existing measures of loneliness (e.g., direct vs. indirect), social isolation (e.g., different scales with different components) and QoL (e.g., frequent use of health-related QoL [13,14] or even health status [10]).

It has been hypothesized that the negative effects of social isolation are mediated by feelings of loneliness, but studies have yielded mixed results, and the need for further research has been highlighted [5,15,16,17]. Moreover, the cultural norms embedded in societies where older people live are an important environmental determinant of loneliness and social isolation, and, therefore, accurate cross-country comparisons are needed [18]. More individualistic societies in Northern Europe have, for decades, reported lower levels of loneliness compared to their counterparts from the Southern Europe, more family-oriented societies [19,20,21]. Still, no prospective studies have focused on identifying and comparing the associations between loneliness/social isolation and QoL in the North and South of Europe.

Our study aimed to determine the relationships between loneliness, social isolation and QoL in older adults; to identify possible North–South Europe differences in these associations and to explore whether the association with social isolation is independent of loneliness.

## 2. Methods

### 2.1. Study Sample

The study sample consisted of 2995 older adults from Sweden and 4154 older adults from Spain, aged 50 years old and more, and followed up for a period of two years between 2015 and 2017, corresponding to waves 6 and 7 of the SHARE study [22,23]. SHARE is an ongoing cross-national, multidisciplinary panel database with information on the health, socioeconomic status and interpersonal networks of adults aged over 50 and their partners across 27 European countries and Israel [24]. Data was collected using computer-assisted personalized interviews (CAPI) at the respondents’ households. The questionnaire is identical in all countries, while the sampling and fieldwork methodology is centrally coordinated. The response rate in wave 7 of those who participated in wave 6 was 77.3% in Sweden and 74.8% in Spain.

### 2.2. Ethics Approval

The study was performed in compliance with the 1964 Declaration of Helsinki and its further amendments. The Study on Health, Aging and Retirement in Europe (SHARE) project is subject to continuous ethical review. The waves used in this study (6 and 7) were approved by the ethics council of Max Planck Society in Munich, Germany, where decisions for all countries are centralized. Previous waves were ethically reviewed by the ethics committee at the University of Mannheim. The study was also approved by the Institute of Health Carlos III ethics committee, CEI PI 62-2019.

### 2.3. Consent to Participate

Informed consent was obtained from all participants in SHARE.

### 2.4. Exposure Variables

Loneliness was measured at wave 6 using the Three-Item Loneliness Scale, which is a short version of the Revised-University of California, Los Angeles (R-UCLA) Loneliness Scale [25]. It is one of the most commonly used measures of loneliness among older adults [6] and has previously been used to make cross-national comparisons in SHARE [26,27,28,29]. The three items of the questionnaire (How much of the time do you feel lack of companionship?, How much of the time do you feel left out? and How much of the time do you feel isolated?) are answered on a three-point Likert-type scale (often, some of the time and hardly ever or never), with the final scale scores ranging from 3 to 9 and higher values reflecting higher loneliness. The continuous variable was standardized (mean of 0 and a standard deviation (SD) of 1) and later dichotomized using a score of ≥5 (highest vs. four lowest quintiles) based on the data distribution given the absence of a validated threshold. The scale showed acceptable internal consistency in both countries (Cronbach’s alpha = 0.65 in Sweden and 0.81 in Spain).

Social isolation was assessed at wave 6 using the social connectedness scale developed by Litwin et al. [30]. This scale is a composite measure that assesses the resources available within one’s social network using the name-based social network module. It takes into account the network size, proximity, contact frequency, support and diversity. Network size is measured by the number of persons cited, proximity by the number of cited social network members living within 25 km, contact frequency by the number of cited persons with weekly or more frequent contact, support by the number of cited persons with very or extremely close emotional ties, and diversity by the number of different types of relationships. Each item has a maximum of 4 points, and the sum is condensed into a calibrated measure using the following conversion: 0 = 0, 1 = 1–5, 2 = 6–10, 3 = 11–15 and 4 = 16–20. The scale was inverted so that higher values reflected higher social isolation. The final scale was standardized and later dichotomized using a score of ≥3 (highest vs. four lowest quintiles), as the literature does not provide a validated threshold.

### 2.5. Outcome Variable: Quality of Life

QoL was assessed at waves 6 and 7 through an adapted version of the CASP-12 index that has shown good psychometric properties and cross-cultural robustness among SHARE participant countries [31]. It is adapted to more advanced ages and consists of 12 items that can be grouped into four dimensions: control, autonomy, self-realization and pleasure. The answers are coded on a 4-point Likert-type scale (often, sometimes, rarely and never). The scoring of the scale ranges from 12 to 48, where high values indicate a better QoL. Only the total score was used in the analyses. The scale showed acceptable internal consistency both at baseline (Cronbach’s alpha = 0.79 in Sweden and 0.82 in Spain) and follow-up (Cronbach’s alpha = 0.78 in Sweden and 0.84 in Spain).

### 2.6. Covariates

The choice of covariates was based on the literature, as well as thorough theoretical considerations by authors [5,30]. Information on age, gender, wealth, education, number of activity limitations, number of chronic conditions, memory, hearing, eyesight and baseline QoL was self-reported in wave 6. The wealth variable was created by summing the values of the main residence, other real estate, bank accounts, bonds, business shares and cars and subtracting the mortgage and financial liabilities [32]. Education was based on the International Standard Classification of Education 1997 (ISCED-1997). Depression was assessed using the EURO-D scale [33]. Memory, hearing and eyesight were self-reported on a 5-point Likert-type scale from “poor” to “excellent”.

### 2.7. Statistical Analyses

Baseline characteristics of the study sample and participants lost to follow-up were compared within each country. Correlations were measured using Pearson’s correlation coefficient (r). Linear regression models were used to assess the longitudinal relationship between the exposures and the outcome. Exposure variables were included both as continuous and binary variables. Baseline CASP was included as a confounder in all models in order to model change in QoL. Selected confounders were added in a step-wise manner; model 1 was adjusted for CASP at baseline; model 2 was adjusted additionally for age, gender, education and wealth; models 3–5 were further adjusted for chronic conditions, depressive symptoms and activity limitations, respectively, and model 6 was adjusted for all covariates. The final model was additionally adjusted for the other exposures (i.e., social isolation in the models on loneliness and vice versa), in order to examine the independent associations of each exposure. Effect modification by age, gender, education, activity limitations, chronic diseases, depression and the other exposures (i.e., loneliness or social isolation, respectively) was assessed by including interaction terms between each of these and the exposures in the models.

Hot-deck imputation and multiple imputation using chained equations (MICE) were used to create five independent imputations of each missing observation in variables with low missingness (<11%). Longitudinal weights were used in all analyses in order to account for a unit nonresponse and make the sample more representative of the target population in each country (i.e., 3,447,937 older adults in Sweden and 16,365,596 older adults in Spain). All statistical analyses were performed using the imputed, weighted dataset and stratified by country in order to perform cross-country comparisons. Estimates from the five imputed datasets were combined using Rubin’s rule [34]. Robust standard errors were obtained to account for intrahousehold cohabitation among participants.

All statistical analyses were performed in StataSE 15 (StataCorp LLC, College Station, TX, USA). Significance level was set at 0.05 (two-tailed test) for all analyses.

## 3. Results

Out of the 9428 participants from Sweden and Spain in wave 6, the 7149 that additionally participated in wave 7 comprised the analytical sample of the study. Those lost to follow-up in Sweden were more likely to have lower education and QoL; higher social isolation and report more activity limitations, chronic conditions, depressive symptoms and cognitive and sensory impairments (all *p* <0.001) (Appendix A). Those lost to follow-up in Spain were more likely to be older and report higher social isolation and more activity limitations, hearing problems and eyesight problems (all *p* <0.05) (Appendix A).

Swedish older adults reported higher education; less activity limitations; chronic diseases and depressive symptoms and better memory, hearing, eyesight and QoL (39.5 vs. 35.6) compared to their Spanish counterparts (Table 1). The mean loneliness score was similar in the two countries (3.74 vs. 3.75), while that for social isolation was higher in Spain compared to Sweden (1.91 vs. 1.82). Unweighted, unimputed baseline characteristics are presented in Appendix A.

In both countries, higher loneliness was significantly associated with older age, female gender, lower wealth and education, poorer physical and mental health and cognitive and sensory impairments, as well as with lower QoL (all *p* <0.01) (Table 2). In Sweden, higher loneliness was additionally associated with higher social isolation (*p* <0.001); however, no such association was observed in Spain. As for social isolation, in Sweden, higher levels were significantly associated with older age, male gender, lower wealth and education, more activity limitations and loneliness and poorer cognition and QoL. In Spain, higher isolation was associated with being male, lower wealth, poorer mental health, hearing and QoL and higher scores on the loneliness scale (all *p* <0.05). Loneliness and social isolation were not highly correlated with one another (r = 0.08 in Sweden and 0.06 in Spain). The baseline QoL was negatively correlated with loneliness (r = −0.49 in Sweden and −0.46 in Spain) and social isolation (r = −0.11 in Sweden and −0.09 in Spain).

In Sweden, loneliness, operationalized as a continuous, standardized variable, was associated with decreased QoL at follow-up (Table 3). However, this association did not reach statistical significance in the fully adjusted model. In contrast, after dichotomizing loneliness, higher levels were significantly associated with a lower CASP-12 score at follow-up in all models, including the one adjusting for social isolation (β: −0.97, 95% CI: −1.55, −0.40). One SD increase in social isolation was associated with a 0.22 (95% CI: -0.38, -0.05)-unit decrease in QoL after controlling for sociodemographic factors. Further adjustments did not attenuate the associations. A statistically significant interaction was found between loneliness and activity limitations (*p* for interaction = 0.02); the association was, in fact, only statistically significant in the group with no activity limitations (β: −0.37, 95% CI: −0.70, −0.03).

In Spain, no significant associations were found between loneliness and QoL once the sociodemographic factors were controlled for (Table 4).

In contrast, every SD increase in social isolation was associated with a lower CASP-12 score by 0.66 points (95% CI: −1.11, −0.22) in the fully adjusted model. The effect size was similar and statistically significant in all models and was not attenuated by feelings of loneliness. The interaction terms of social isolation with age and chronic diseases were close to statistical significance (*p* = 0.07 for both). After stratifying by age (≤ and >65 years), a stronger association was observed in the younger age group, for whom every standard deviation increase in social isolation was associated with a 1.16-point decrease in the CASP-12 score (95% CI: −1.94, −0.38) in the fully adjusted model (Figure 1A). As for the number of chronic diseases, a stronger association was observed in the group with no chronic diseases (β = −1.40, 95% CI: −2.44, −0.37 vs. β = −0.46, 95% CI: −0.82, -0.10) (Figure 1B). 

## 4. Discussion

Our study results showed that loneliness was associated with QoL in Swedish older adults, even if the association was limited to higher levels of loneliness. In contrast, the negative association of social isolation seemed to be more uniformly distributed across the whole range of the exposure. In Spain, loneliness was not significantly associated with QoL, while social isolation was. Further stratification revealed that the association was noticeably stronger in those under the age of 65 and those with no self-reported chronic diseases. No such effect modification was seen in Sweden, where the association was similar in both age groups. The association between social isolation and QoL was independent of loneliness in both countries.

The fact that we did not find any differences in the baseline prevalence of loneliness and social isolation between both countries departs from the previous evidence arising from Europe, which has consistently shown the highest rates of loneliness in Mediterranean countries and the lowest rates in Nordic countries [19,20,21]. The analysis of the characteristics of participants lost to follow-up did not reveal differences in the loneliness scores. However, it is possible that nonresponse in previous waves of SHARE was differential in regard to loneliness, with lonely Spanish individuals being more likely to drop out of the study. Discrepancies among findings could also be due to aspects related to the assessment of loneliness; while our study used an indirect measure whereby any explicit reference to loneliness is avoided, the studies that did find a North–South gradient in Europe were based on direct measures. According to Shiovitz-Ezra et al., indirect and direct measures of loneliness do not have a high degree of concordance, and the latter are more likely to classify an individual as lonely [35]. The authors also found that the two measures had different correlates; for example, older adults were more likely to report loneliness using the direct scale, while more educated individuals were less likely to do so based on indirect measures. It is therefore possible that older adults in Sweden report loneliness differently on the indirect scale compared to their Spanish counterparts.

Our study found a negative association between higher loneliness and QoL among older individuals, albeit only in Sweden. The strong association remained even after adjusting for depression, one of the major mediators of the negative health effects associated with loneliness [36]. A longitudinal study of a nationally representative cohort of older adults in Ireland using the same measure of QoL as ours found that multiple social factors, including loneliness, had a strong impact on QoL at follow-up [37]. Feelings of social connectedness may operate through different psychosocial mechanisms other than mental health to increase older people’s QoL, such as fulfilling the need to belong [38] and providing a sense of control [39], as well as buffering biological stress processes [40]. The reason why no association between loneliness and QoL was found in Spain remains to be elucidated. If we hold the aforementioned well-documented European North–South gradient in loneliness to still be true today, this would signal an underestimation of the prevalence of loneliness in Spain, which could explain the lack of association, particularly if it is limited to higher levels of loneliness as seen in the Swedish sample.

Another important finding of the study is the negative relationship observed between social isolation and QoL in both countries. It is suggested that, by meaningfully interacting with other people, older adults may improve their health-related behaviors, nutrition, physical activity and healthcare accessibility. These changes are likely to impact the control and autonomy components of QoL, while the psychological and stress-buffering benefits of having a strong and diverse social network [41] may be especially linked to an improvement in the pleasure and self-realization components. Our results are in line with previous longitudinal studies from Eastern Europe and the United Kingdom [42,43]. Additionally, a cross-sectional study by Litwin et al. also found a positive association of social connectedness with QoL using SHARE data, after adjusting for similar covariates as those in our study [30].

The strong interaction between social isolation and age found in Spain could possibly be explained by the socioemotional selectivity theory. Proposed by Carstensen in 1999 [44], it stipulates that the perception of time has a strong influence on the type of relationships humans pursue. Those who view the future as open-ended are more likely to seek out knowledge connections and have a wider social network, while those who view it as finite are more likely to focus on the emotional components of their relationships and not on objective factors such as size. Thus, younger older adults may be especially affected by a suboptimal social network in terms of size and connections, as is the case in Spain. Indeed, a meta-analysis found a stronger association between social isolation and mortality in studies with a participant mean age below 65 years [7]. The absence of an age interaction in Sweden may be attributed to differences in societal structures and roles between countries. In nonindividualistic countries such as Spain, strong ties with the family and community are kept for longer, and perhaps, the transition process happens around the age of 65. In the more individualistic context of countries such as Sweden, said transition process could be happening earlier, around or before the age of 50—in which case, it would not be captured in our study. Considering the interactions with a number of chronic diseases, another possible explanation could be that the negative effects of social isolation are moderated by health status, i.e., biologically younger older adults are the ones who may benefit most from being socially connected.

Lastly, our finding that loneliness did not alter the association between social isolation and QoL is in line with previous findings that suggest a low correlation between the two phenomena and independent pathways for both [5,15]. The two phenomena might diverge even further in old age, as indicated by the social selectivity theory. Indeed, previous research has suggested that loneliness and social isolation have their own, distinct pathways through which they affect individuals’ health [4].

### 4.1. Strengths and Limitations

A major strength of the study is the use of an ex-ante harmonized, longitudinal and multi-disciplinary survey. SHARE provides a large, representative, population-based cohort and allows for controlling for several health, demographic and socioeconomic factors; it is also specifically designed to make cross-country comparisons in Europe. The ex-ante harmonization guarantees that both the interview process, as well as the fieldwork, are identical among participating countries [45]. This greatly enhances the validity of our comparisons by minimizing the artefacts that may arise from country-specific survey designs. Further strengths are the use of a validated scale to assess loneliness and a scale designed specifically for SHARE to assess social isolation, thus maximizing the use of information available in the questionnaire. The latter scale takes into account the subjectivity of personal social networks and makes no a priori assumptions about the types of relationships that are most important to older adults (e.g., family and friends), which is a common practice in the field due to practical reasons. Instead, respondents themselves provide names of the most important people in their lives and answer subsequent questions about their relationships with these specific people. Social relationships are referred to as double-edged swords, with the same objective relationship (e.g., child or sibling) having the ability to be both protective and threatening [46]. Thus, the name-based method enhances the internal validity of the study by minimizing misclassification bias.

Some limitations of the study need to be taken into consideration. Selection bias is a common threat in epidemiological studies, and our study is no exception. The initial samples from SHARE wave 1 were replenished in wave 4 in Spain and wave 5 in Sweden, meaning that our baseline sample at wave 6 already suffered from nonresponses and attrition in at least two previous waves. While those who were lost to follow-up were significantly different from those who were not across certain characteristics, the absolute differences were not too big. The problem was partly addressed by using the calibrated longitudinal weights in all analyses. Since the weights were provided for the entire longitudinal sample, we did not exclude those living in nursing homes or presenting cognitive impairments; however, the number of participants living in retirement homes was low, and those with cognitive problems were interviewed using proxies, so the results were not likely to change.

Another potential limitation is the short period of follow-up, which could prevent us from detecting significant changes in the CASP-12 scores. In spite of this, we found statistically significant associations for both loneliness and social isolation. To the best of our knowledge, the literature does not provide a clinically relevant threshold for CASP-12 or for minimal clinically important changes, so it is difficult to put the changes into perspective. However, it is expected that even small changes in QoL may have relevant consequences, especially in older ages. Due to the exposures being measured in close temporal proximity to the outcome, the possibility of reverse causation cannot be excluded. Studies with longer follow-up periods are needed to confirm the direction of the association.

### 4.2. Public Health Implications and Future Research Considerations

Healthcare professionals and social workers, who are ideally positioned to identify older adults at risk of loneliness and social isolation, may increase their efforts to promote and convey the importance of social connectedness among their older patients as a means of healthy aging. Policymakers should consider that these phenomena are contextually bound and carefully evaluate and adapt the interventions to their respective settings. Interventions should possibly be upscaled from the individual and group levels to the population level, based on previous evidence addressing exposures of similar natures [47,48]. Still, more epidemiological research is needed based on the subsequent waves of SHARE, as well as other European national surveys, to replicate these findings and explore the mechanisms underlying these associations.

## 5. Conclusions

Loneliness and social isolation are prospectively associated with lower QoL in older adulthood, but the associations are contextually bound and, thus, slightly differ between countries. The association between social isolation and QoL is independent of feelings of loneliness. In light of the rapid population aging in Western societies, loneliness and social isolation may be relevant modifiable factors to target in order to ensure healthy aging.

## Figures and Tables

**Figure 1 ijerph-17-08637-f001:**
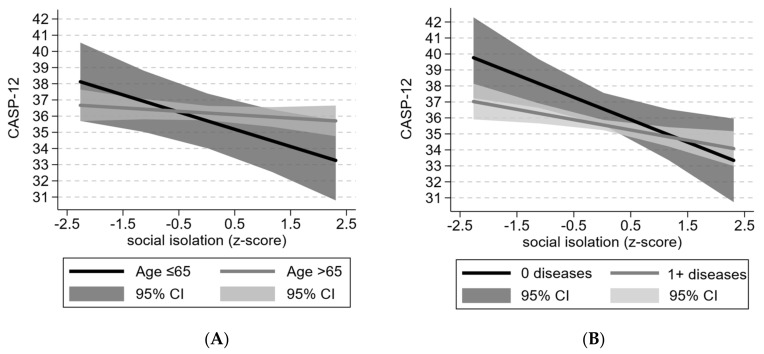
Predicted CASP-12 score at follow-up by levels of social isolation in Spain stratified by (**A**) age and (**B**) number of chronic diseases. Results based on fully adjusted models.

**Table 1 ijerph-17-08637-t001:** Baseline characteristics (means, percentages and 95% CI) of weighted sample by country.

	Sweden	Spain
	*n* = 3,447,937	*n* = 16,364,596
**Age**	66.8 (66.2–67.3)	66.5 (64.7–68.2)
50–64	46.6% (44.1–49.0)	48.9% (41.1–56.8)
65–74	31.0% (29.2–32.8)	25.8% (21.9–29.7)
75–84	16.7% (15.3–18.0)	18.4% (15.3–21.4)
85+	5.8% (4.9–6.7)	6.9% (5.3–8.6)
**Gender**		
Male	47.9% (45.5–50.2)	46.0% (43.8–48.2)
Female	52.1% (49.8–54.5)	54.0% (51.8–56.2)
**Education level**		
Primary	16.1% (14.6–17.4)	53.7% (48.4–59.0)
Secondary	47.8% (45.5–50.2)	35.6% (30.9–40.2)
Tertiary	36.1% (33.8–38.5)	10.7% (8.1–13.4)
**Activity limitations**		
0	85.3% (83.7–86.8)	82.2% (79.4–84.8)
1+	14.7% (13.2–16.3)	17.8% (15.2–20.6)
**Chronic diseases**	1.31 (1.25–1.37)	1.86 (1.71–2.00)
0	34.3% (31.9–36.7)	21.8% (17.8–25.8)
1	30.4% (28.3–32.6)	26.7% (23.2–30.3)
2+	35.3% (33.1–37.5)	51.5% (46.6–56.4)
**EURO-D**	1.98 (1.89–2.10)	2.34 (2.12–2.55)
<4	81.8% (80.0–83.7)	73.3% (70.0–76.5)
≥4	18.2% (16.3–20-1)	26.7% (23.5–30.0)
**Memory**		
Fair/Poor	23.3% (11.7–15-2)	38.2% (34.6–41.9)
Excellent/Very good/Good	76.7% (74.8–78.6)	61.8% (58.1–65.4)
**Hearing**		
Fair/Poor	17.0% (15.3–18.7)	21.4% (18.2–24.6)
Excellent/Very good/Good	83.0% (81.2–84.7)	78.6% (75.4–81.8)
**Eyesight**		
Fair/Poor	13.5% (11.7–15-2)	21.9% (19.1–24.6)
Excellent/Very good/Good	86.5% (84–8-88.3)	78.1% (75.4–81.0)
**Loneliness**	3.74 (3.68–3.80)	3.75 (3.66–3.85)
Low	81.9% (80.1–83.4)	79.8% (77.1–82.6)
High	18.1% (16.2–20.0)	20.2% (17.4–22.9)
**Social Isolation**	1.82 (1.78–1.87)	1.91 (1.83–2.01)
Low	80.0% (78.1–81.9)	75.0% (71.0–79.1)
High	20.0% (18.1–21.9)	25.0% (20.1–29.0)
**CASP-12 (baseline)**	39.5 (39.2–39.8)	35.6 (35.1–36.1)

**Table 2 ijerph-17-08637-t002:** Baseline characteristics (means and percentages) of the weighted samples by exposure status and country.

	Sweden						Spain					
	Loneliness			Isolation			Loneliness			Isolation		
	Low	High	*p*-Value	Low	High	*p*-Value	Low	High	*p*-Value	Low	High	*p*-Value
***n***	2470	525		2366	629		3329	825		2772	1382	
**Age**	66.1	69.9	<0.001	66.3	68.8	<0.001	65.3	71.0	<0.001	66.2	67.2	0.32
50–64	48.9%	36.4%		48.7%	37.8%		53.1%	32.3%		50.0%	45.8%	
65–74	31.3%	29.6%		30.6%	32.8%		26.0%	25.1%		25.0%	28.1%	
75–84	15.5%	21.3%		15.7%	20.4%		15.7%	28.9%		18.8%	17.0%	
85+	4.3%	12.7%		5.0%	9.0%		5.2%	13.7%		6.2%	9.1%	
**Gender (female)**	49.4%	64.4%	<0.001	56.0%	36.7%	<0.001	50.0%	69.0%	<0.001	57.1%	44.8%	0.001
**Wealth (thousand €)**	44.6	27.1	<0.001	43.2	34.2	0.013	22.6	15.0	0.002	22.6	16.6	0.01
**Education level**			<0.001			<0.001			0.001			0.62
Primary	13.6%	27.1%		14.1%	24.0%		49.8%	68.8%		53.9%	53.1%	
Secondary	48.2%	46.1%		47.0%	51.3%		38.8%	23.0%		34.5%	38.7%	
Tertiary	38.2%	26.8%		38.9%	24.7%		11.4%	8.2%		11.6%	8.2%	
**Activity limitations**			<0.001			0.08			<0.001			0.25
0	88.6%	70.0%		85.8%	83.1%		86.4%	65.2%		83.1%	79.0%	
1+	11.4%	30.0%		14.2%	16.9%		13.6%	34.8%		16.9%	21.0%	
**Chronic diseases**	1.21	1.75	<0.001	1.32	1.28	0.60	1.69	2.52	<0.001	1.80	2.00	0.13
0	37.1%	21.6%		34.1%	35.1%		24.7%	10.3%		22.9%	18.4%	
1	31.0%	27.8%		30.6%	29.5%		28.1%	21.3%		26.3%	28.0%	
2+	31.9%	50.6%		35.3%	35.4%		47.2%	68.4%		50.8%	53.6%	
**EURO-D**	1.67	3.40	<0.001	1.99	1.96	0.85	1.74	4.68	<0.001	2.23	2.66	0.03
<4	86.3%	61.4%		81.3%	84.0%		82.0%	38.5%		74.9%	68.4%	
≥4	13.7%	38.6%		18.7%	16.0%		18.0%	61.5%		25.1%	31.6%	
**Memory (fair/poor)**	20.9%	34.2%	<0.001	22.2%	27.7%	<0.001	33.0%	58.9%	<0.001	37.9%	39.3%	0.56
**Hearing (fair/poor)**	15.8%	22.6%	0.001	16.7%	18.7%	0.30	18.6%	32.3%	0.001	18.8%	29.2%	0.01
**Eyesight (fair/poor)**	12.7%	17.0%	<0.001	12.8%	15.9%	0.14	19.4%	31.7%	0.001	22.3%	20.5%	0.23
**CASP-12 (baseline)**	40.6	34.7	<0.001	39.8	38.4	<0.001	36.8	30.8	0.001	36.3	33.3	<0.001
**Loneliness**	-	-	-	3.68	3.93	0.002	-	-	-	3.70	3.91	0.03
Low	-	-	-	83.3%	76.6%		-	-	-	80.5%	77.8%	
High	-	-	-	16.7%	23.4%		-	-	-	19.5%	22.2%	
**Social isolation**	1.78	2.03	<0.001	-	-	-	1.91	1.95	0.50	-	-	-
Low	81.3%	74.1%		-	-	-	75.7%	72.6%		-	-	-
High	18.7%	25.9%		-	-	-	24.3%	27.4%		-	-	-

Note: *p*-values obtained using ANOVA. Percentages obtained from one imputation dataset.

**Table 3 ijerph-17-08637-t003:** Linear regression coefficients (95% CI) for the CASP-12 score at follow-up in Sweden.

			Loneliness				Social Isolation		
Model	Variables Included	Continuous	*p*-Value	Binary	*p*-Value	Continuous	*p*-Value	Binary	*p*-Value
1	Crude	−0.43 (−0.74, −0.12)	0.007	−1.33 (−1.91, −0.76)	<0.001	−0.31 (−0.46, −0.15)	<0.001	−0.31 (−0.74, 0.11)	0.147
2	Sociodemographic factors ^a^	−0.37 (−0.66, −0.07)	0.016	−1.15 (−1.72, −0.57)	<0.001	−0.22 (−0.38, −0.05)	0.009	−0.14 (−0.55, 0.28)	0.512
3	Model 2 + chronic diseases	−0.35 (−0.64, −0.05)	0.02	−1.13 (−1.70, −0.56)	<0.001	−0.25 (−0.41, −0.08)	0.003	−0.20 (−0.61, 0.21)	0.346
4	Model 2 + EURO-D	−0.29 (−0.59, 0.01)	0.06	−1.00 (−1.59, −0.41)	<0.001	−0.23 (−0.39, −0.06)	0.007	−0.17 (−0.58, 0.24)	0.420
5	Model 2 + activity limitations	−0.32 (−0.62, −0.02)	0.039	−1.10 (−1.68, −0.52)	<0.001	−0.22 (−0.39, −0.06)	0.007	−0.13 (−0.55, 0.28)	0.531
6	Fully adjusted model ^b^	−0.27 (−0.56, 0.02)	0.072	−1.01 (−1.60, −0.43)	<0.001	−0.27 (−0.43, −0.10)	0.002	−0.21 (−0.62, 0.20)	0.308
7	Model 6 + other exposure ^c^	−0.25 (−0.54, 0.04)	0.096	−0.97 (−1.55, −0.40)	<0.001	−0.25 (−0.42, −0.09)	0.003	−0.19 (−0.60, 0.22)	0.358

^a^ Adjusted for the baseline CASP-12 score, age, gender, wealth and education; ^b^ adjusted for the baseline CASP-12 score, age, gender, wealth, education, activity limitations, EURO-D, chronic diseases, memory, hearing and eyesight and ^c^ social isolation in the models on loneliness and vice versa.

**Table 4 ijerph-17-08637-t004:** Linear regression coefficients (95% CI) for the CASP-12 score at follow-up in Spain.

			Loneliness				Social Isolation		
Model	Variables Included	Continuous	*p*-Value	Binary	*p*-Value	Continuous	*p*-Value	Binary	*p*-Value
1	Crude	−0.41 (−0.79, −0.03)	0.038	−0.98 (−2.07, 0.10)	.075	−0.54 (−1.06, 0.01)	0.045	−0.18 (−1.32, 0.95)	0.742
2	Sociodemographic factors ^a^	−0.16 (−0.53, 0.21)	0.399	−0.35 (−1.32, 0.61)	.474	−0.60 (−1.08, −0.13)	0.014	−0.27 (−1.28, 0.74)	0.592
3	Model 2 + chronic diseases	−0.15 (−0.53, −0.23)	0.436	−0.33 (−1.29, 0.64)	.508	−0.60 (−1.06, −0.13)	0.012	−0.29 (−1.28, 0.71)	0.566
4	Model 2 + EURO-D	0.15 (−0.22, 0.53)	0.423	−0.34 (−0.66, 1.35)	.504	−0.62 (−1.08, −0.17)	0.008	−0.32 (−1.30, 0.65)	0.507
5	Model 2 + activity limitations	−0.16 (−0.53, 0.21)	0.392	−0.42 (−1.38, 0.54)	.391	−0.63 (−1.11, −0.16)	0.009	−0.33 (−1.33, 0.68)	0.513
6	Fully adjusted model ^b^	0.13 (−0.26, 0.51)	0.515	0.27 (−0.73, 1.28)	.596	−0.66 (−1.11, −0.22)	0.004	−0.43 (−1.39, 0.54)	0.379
7	Model 6 + other exposure ^c^	0.14 (−0.24, 0.51)	0.476	0.23 (−0.74, 1.19)	.643	−0.66 (−1.11, −0.22)	0.004	−0.43 (−1.39, 0.54)	0.379

^a^ Adjusted for the baseline CASP-12 score, age, gender, wealth and education; ^b^ adjusted for the baseline CASP-12 score, age, gender, wealth, education, activity limitations, EURO-D, chronic diseases, memory, hearing and eyesight and ^c^ social isolation in the models on loneliness and vice versa.

## Data Availability

Data supporting our findings is available to the scientific community free of charge at www.share-project.org (Wave 6—10.6103/SHARE.w6.710 and Wave 7—10.6103/SHARE.w7.710).

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
