# Peer review of "Are Loneliness and Social Isolation Associated with Quality of Life in Older Adults? Insights from Northern and Southern Europe"

_ijerph, 2020, doi:10.3390/ijerph17228637_

Round 1

Reviewer 1 Report

The paper is of high scientific interest, addressing an increasingly relevant social issue: quality of life in old age. It draws on waves 6 and 7 of the SHARE study with an impressive number of participants.

One of its main interests consists in the comparison that is made between a Southern and a Northern European country. Regarding this aspect, however, the study would gain considerable consistency if at least the Loneliness Scale and the CASP-12 were subject to Confirmatory Factor Analysis (CFA) and a test of Factorial Invariance (between the two groups under study: the Swedish and the Spanish participants). Only through a CFA and the invariance testing would we be sure that the scales are being interpreted in a similar fashion by the two groups, and that ultimately one would be comparing the same constructs. Furthermore, a measure of internal consistency would also be required.

The authors, mention the fact that CASP-12 has shown good psychometric properties in different countries. They also mention an ex-ante harmonization and the use of validated scales. Yet, what should be of concern for the present study is whether this particular study's data actually fit the scales' factorial models or not. And for that an CFA and its group invariance testing would be advisable. 

Regarding the Social Connectedness Scale, one may argue that it refers to aspects that are being more objectively measured and not so subject to the participants interpretation. So, with regard to this scale the need for a CFA is not so strong.

The interesting results and their discussion would gain in robustness should the scales be subject to CFA, invariance testing and internal consistency values presented.

Reviewer 2 Report

Although the study is interesting with some interesting results, the paper is difficult to read.

  1. The relationship between social isolation is discussed well in the introduction as relatively independent constructs—it would be helpful to know the correlation between the two variables. It seems that the researchers have these data and can provide such a result. Do both variables predict the same health outcomes in prior studies? And how each of these correlates with QOL. The various complex analyses presented do not get at these simple questions.
  2. The authors use the expression Study Population—if they are studying the entire population then why do we need statistical tests of significance? They do have large numbers—but are these samples or populations? Later, in the Results section, they use the word Sample, and also differentiate Respondents from Non-Respondents, but the latter was not defined in the Method section. How did they get the data from or about Non-Respondents? What are Non-Respondents?

Table numbering is confusing. Table 1, Supplementary Table 1 why not just use Tables 1, 2, etc.  Supplementary Table 3 comes after Table 1 and not after Table 3. Also, in Table the N = 3,44,7937 and 16, 364, 596—these large numbers are included without explanation, where did the data for these numbers come from? These numbers are not the same as those given in the Study Population section. The numbers keep changing and hard to follow was where they are coming from, for example, 9,428 & 7,149. 

  1. More details are needed on the variables included. Give there are 3 items of the Loneliness scale, include all items, and indicate its reliability found in your study. The authors mention skewness for only the Social Isolation measure, but not for other measures included in their study. Why not also report Kurtosis? Was there an index computed, what were the values? No details on the reliability of the QOL score obtained in the current study are given. There are 4 subscales—did they just use the total score or the analysis was done separately for each subscale. The same comments can be applied to some variables included as covariates—for example, depression, what is its reliability?. How was memory measured? In the results section the authors mention cognitive and sensory impairment—but there is no mention of how they were measured.
  2. There should be some rationale presented for including so many covariates.
  3. In some Tables, titles say proportions, but percentages are included in the body of the table.
  4. Also, p values are reported as 0.00, 0.05, etc. Since p values cannot exceed 1.0, report them as .00, .01, .05. Also, why not report exact p values obtained, instead of < where possible? For CIs why not just one alpha value instead of using different values? Using one alpha value will make results more comparable.
  5. In Discussion the authors talk about the “effects of social isolation”–since this is a correlational study, it would be better to avoid causal implications.

Reviewer 3 Report

Review of the manuscript: Are loneliness and social isolation associated with quality of life in older adults? Insights from Northern and Southern Europe
This report presented a really interesting set of findings based on data derived from participants from 2 different countries. The data the authors had collected (for a different purpose/s) indicated that concepts of loneliness, social isolation, and quality of life are intricately related depending on the cultural background/backdrop of its participants. The report is well-written with good analyses and sound discussion.
There are a few considerations or suggested revisions I ask:
1. For a reader who is unfamiliar with the original data set, where there were
apparently several “Waves” of data collection, it was unclear what that original
study was about. If the authors could talk more about that in general, with
enough information so the readers will have some idea what the general
database looks like from which the relevant data were analyzed for this particular focus of analyses, it would be a bit clearer.
2. It would also be clearer if the assessment of the relevant variables of loneliness, social isolation, and quality of life were originally intended to be measured in the original purpose of the study or if this set of hypothesized relationships between the variables in the 2 countries were post hoc hypotheses.
3. Regarding the questionnaire used to measure loneliness (i.e., the three-item
loneliness scale), why was this specific scale chosen? The scale only had 3
items in it with the response format of a Likert scale ranging in 3 points only. It
seems very lacking in psychometrical robustness and soundness. What were
the psychometric values of the questionnaire? Example, what was its Cronbach’s alpha? Who used and more importantly, promoted this short scale as a valid scale to use in any study? There are other loneliness scales with more than 3 items in it that have a better range of response format. Why were they not used instead?
4. Regarding the assessment questionnaires used in the study, was it necessary to translate the items to a different language? Was it necessary to check the potential misunderstanding of the meaning of some items in the questionnaires because of cultural differences in perspectives? What were some of the steps taken to control for or to handle cultural diversity in the interpretation of the items in the questionnaires?

Round 2

Reviewer 1 Report

The text has been subject to improvement: Important corrections were made throughout the text; New data were introduced (internal consistency values of the scales used).

Given the corrections and the new information reported, as well as the fact that SHARE is an ongoing project with already multiple waves of data collection, I believe it's reasonable to rely on the good psychometric properties and cross-cultural robustness of previous waves among SHARE participant countries. Taking this into account I will not insist with the performance of Confirmatory Factor Analyses of the scales that were used in the study. Yet, as the authors point out in the discussion for the loneliness measure, it is "possible that older adults in Sweden report loneliness differently on the indirect scale compared to their Spanish counterparts". A Confirmatory Factor Analysis with invariance testing would have clarified this aspect. As it was not performed, one has to rely on past "good psychometric properties (as a proxy for the non-existance of psychometric properties of the scales in the present paper). Results must therefore be interpreted with extreme caution, as the authors seem to be aware when they express the concern quoted above.